# Transcriptome Analysis of Two Tetraploid Potato Varieties under Water-Stress Conditions

**DOI:** 10.3390/ijms232213905

**Published:** 2022-11-11

**Authors:** Alba Alvarez-Morezuelas, Leire Barandalla, Enrique Ritter, Jose Ignacio Ruiz de Galarreta

**Affiliations:** NEIKER-Basque Institute for Agricultural Research and Development, Basque Research and Technology Alliance (BRTA), Campus Agroalimentario de Arkaute, 01192 Arkaute, Spain

**Keywords:** drought stress, *Solanum tuberosum* L., RNAseq, differentially expressed genes

## Abstract

Potato (*Solanum tuberosum* L.) is one of the most important crops worldwide, but due to its sensitivity to drought, its production can be affected by water availability. In this study, the varieties Agria and Zorba were used to determine the expression differences between control and water-stressed plants. For this purpose, they were sequenced by RNAseq, obtaining around 50 million transcripts for each variety and treatment. When comparing the significant transcripts obtained from control and drought-stressed plants of the Agria variety, we detected 931 genes that were upregulated and 2077 genes that were downregulated under stress conditions. When both treatments were compared in Zorba plants, 735 genes were found to be upregulated and 923 genes were found to be downregulated. Significantly more DEGs were found in the Agria variety, indicating a good stress response of this variety. “Abscisic acid and environmental stress-inducible protein TAS14-like” was the most overexpressed gene under drought conditions in both varieties, but expression differences were also found in numerous transcription factors and heat shock proteins. The principal GO term found was “cellular components”, more specifically related to the cell membrane and the cell wall, but other metabolic pathways such as carbohydrate metabolism and osmotic adjustment were also identified. These results provide valuable information related to the molecular mechanisms of tolerance to water stress in order to establish the basis for breeding new, more tolerant varieties.

## 1. Introduction

Over the last few decades, there has been a global increase in temperatures, resulting in an increase in the frequency and severity of drought conditions [1]. This situation is expected to worsen in the coming decades and will even require irrigation in traditionally rain-fed areas [2]. Drought leads to important reductions in crop yields, particularly in potato, where water stress is the second cause of yield loss after pathogens [3].

Potato crops have relatively high water efficiency, but at the same time have a high water requirement, which makes them very susceptible to water stress [2]. This susceptibility is due to the fact that the crop has a shallow and sparse root system, only about 30–100 cm deep, leading to decreased yield under drought conditions. Breeding for drought tolerance is a critical issue to avoid yield losses [4]. Lack of water affects the crop at all stages of development, from emergence to tuber filling, and has negative effects on parameters such as biomass, yield, tuber number and quality [5,6].

Plants use different strategies to mitigate the effects of water stress by reducing water loss through stomata closure, increasing water absorption from the soil by developing their root system or accelerating their growing cycle. These mechanisms require molecular signaling processes which include transcription factors, protein kinases and stress-related proteins [7].

To mitigate the impact of abiotic stresses on crops, both agronomic and genotypic solutions are needed. Water, soil and plant management have an important influence on the effects of water stress, but in many cases, these methods are not sufficient. It is necessary to combine these techniques with the breeding of varieties adapted to climatic conditions [8]. Potato breeding is a complex and lengthy process, since the crop is autotetraploid, highly heterozygous and inbreeding-depressed [9]. In recent years, new technologies have been developed for studying the genome which can accelerate this process. However, drought tolerance is a complex polygenic trait, which hampers the detection of differentially expressed genes [10]. Transcriptome analysis has been performed in different crops such as wheat [11], grapevine [12], maize [13] or rice [14] for exploring the molecular regulatory mechanisms of plants in response to water stress.

Classical plant breeding methods are based on a phenotypic approach, which is slow, labor-intensive and expensive. Thanks to the increased knowledge of the molecular mechanisms in plants, the transcriptome and the expression patterns of studied genes, plant breeding has been improving in recent years [15]. With the development of high-throughput sequencing technologies, a significant coverage of cDNA sequences from RNA samples can be obtained. By applying these sequencing technologies to different cDNA libraries from samples of interest, transcriptome overviews for profiling can be obtained [16].

The application of next-generation sequencing techniques such as RNAseq allows the generation of gene expression profiles to characterize stress response and provides information for the discovery of new genes and the analysis of metabolic pathways associated with the response to environmental stresses [9,17]. Absolute measurement of gene expression using RNAseq provides quantitative and qualitative information which is more accurate than other previously used techniques such as microarrays [18].

RNAseq assays in potato, targeting diseases such as *Phytophthora infestans* [19] or nematodes [20], salt stress [21], anthocyanin accumulation in tubers [22] and nitrogen response [23], were performed.

In this study, we used this new sequencing technique for analyzing in two varieties the genes that are overexpressed or repressed in order to understand the mechanisms of drought-stress response for accelerating the selection processes in potato breeding programs.

## 2. Results

RNA quality control revealed that all the samples had 28S:18S ratios in a range of 1.8–2.0 and a mean RNA integrity number (RIN) > 7.0, which met the requirements for library construction and sequencing. To obtain a global view of the transcriptome of the potato drought tolerance, four libraries were generated and indexed using mRNA extracted from drought treatment and unstressed control leaves of two varieties. High-throughput RNA sequencing using Illumina technology was performed.

After sequencing and removing low-quality reads, a total of 54,042,913 and 56,473,479 reads were acquired from the drought treatment and control leaves for the Agria variety, and 53,513,496 in the drought treatment and 55,798,228 in control leaves for Zorba variety (Table 1). The GC content was around 43% in all cases, and the majority of reads (around 90%) could be mapped to the reference genome sequence (Table 1).

When control plants were compared with drought ones in the Agria variety, a total of 3008 DEGs were identified, of which 931 were upregulated and 2077 were downregulated. When comparing both treatments in Zorba plants, 735 genes were found to be upregulated and 923 genes were downregulated. The number of up- and downregulated genes was based on |log2 fold change| ≥ 2 and adjusted *p*-value < 0.05 (Figure 1). In both varieties, most of the DEGs were downregulated, indicating that the transcriptomic response to drought was primarily downregulation in gene expression.

When comparing the total number of DEGs, 1277 common DEGs were identified: 510 of these DEGs were upregulated, 755 were downregulated and 12 were oppositely regulated (Figure 2).

The top 10 upregulated genes and downregulated genes are detailed in Table 2 for the Agria variety. A calcium-binding protein, hyoscyamine 6-dioxygenase, L-ascorbate oxidase, GDSL esterase/lipase, abscisic stress-ripening protein and bZIP transcription factor, among others, differed significantly between control and stressed plants. When comparing the differentially expressed genes of the Zorba variety, we detected beta-hexosaminidase, glucan endo-1,3-beta-glucosidase, tropinone reductase, peroxidase 51, O-acyltransferase WSD1 and nonspecific lipid transfer protein differed significantly (Table 3). The most overexpressed gene in both varieties was “abscisic acid and environmental stress-inducible protein TAS14”, with a log2 fold change of 3449.52 in the Agria variety and 692.09 in the Zorba variety. One of the genes that most decreased its expression under water-stress conditions was “36.4 kDa proline-rich protein”, with a log2 fold change of −39.22 and −15.42, respectively.

For obtaining an overview of the putative functions of genes that participate in drought-stress response, GO enrichment analysis was used to identify DEGs between irrigated and nonirrigated plants. The DEGs were classified into three categories: biological processes (BP), cellular components (CC) and molecular function (MF). For the biological process category, “DNA replication initiation” was the only GO term in both varieties (Figure 3 and Figure 4). Eleven GO terms were found for “cellular components” in both varieties, and nine of them were identical: MCM complex, plant-type cell wall, cell wall, external encapsulating structure, plasma membrane, cell periphery, integral component of membrane, intrinsic component of membrane and membrane. Photosystem and apoplast-related DEGs were found in Agria (Figure 3), whereas THO complex and extracellular region-related DEGs were found in Zorba (Figure 4). Eleven GO terms were involved in “molecular function” in Agria and six GO terms in Zorba. Four of them were identical: hydrolase activity hydrolyzing O-glycosyl compounds, hydrolase activity acting on glycosyl bonds, oxidoreductase activity and catalytic activity. In both cases, genes related to catalytic activity were the most abundant, with a total of 703 unique DEGs in Agria and 413 unique DEGs in Zorba. Membrane-related genes also have a very important influence and were the most abundant in the cellular component category. In this case, 540 unique DEGs were found in Agria and 294 unique DEGs in Zorba.

Ten DEGs were randomly selected for RT-qPCR analysis to validate the results of RNA sequencing. The expression levels determined by RT-qPCR followed the same trends as in transcriptome sequencing. In the comparison of control and stressed plants, the correlation coefficients of gene expression trends in sequencing data and RT-qPCR results were 0.952 for Agria and 0.972 for Zorba, indicating that our transcriptome sequencing data were highly reliable (Figure 5).

## 3. Discussion

We have applied in this study mRNA sequencing to evaluate transcriptome changes in potato leaves under water-stress conditions and unstressed control conditions in two genotypes. Using Illumina sequencing technology, we generated around 55 million transcripts in each library. The numbers of differentially expressed genes indicate that there was stress in leaves of the plants under drought stress, with a total of 3008 DEGs in the Agria variety and 1658 in the Zorba variety.

Drought tolerance is a complex trait and involves multiple mechanisms that can act in combination to avoid or tolerate periods of water deficit. Gene expression experiments comparing water-stressed and nonstressed potato plants have been performed in several studies [24,25,26]. As mentioned by [27], plants vary in the timing and speed of response to drought conditions, depending on their genetic background and ecotype, but some drought response genes, such as those involved in osmotic adjustment or cell signaling and communication, are conserved among plant taxa.

The response to water stress is a complex character that exhibits itself in different ways, as evidenced by the large amount of GO enrichment in stressed leaves. In this study, DEG enrichment was observed in the membrane-related category. The cell membrane is one of the main components affected by water stress and is affected in its composition of both phospholipids and proteins that help to maintain membrane integrity, preserve cellular compartments and activate phospholipid signaling pathways in response to stress [28]. In both varieties, we have observed a significant increase in membrane-related genes, such as MCM complex, plant-type cell wall, cell wall, external encapsulating structure, plasma membrane, cell periphery, integral component of membrane, intrinsic component of membrane and membrane, suggesting that these plants have activated water stress defense mechanisms to maintain intracellular water.

In our data, ABA-related genes were significantly expressed in both varieties under drought-stress conditions. Abscisic acid plays an important role in plant adaptation to environmental stresses such as water limitation. Genes involved in ABA biosynthesis, catabolism and signaling represent interesting candidate genes for the breeding of drought-tolerant crops. When plants are under water stress, one of the first responses is the expression of ABA-responsive genes with a consequent increase in this hormone. In their article, [29] mention some possible candidate genes related to abscisic acid biosynthesis, catabolism and signaling, such as abscisic acid receptor (PYL), protein phosphatase 2C (PP2C), or serine/threonine protein kinase (SnRK2). According to [30], under drought stress, ABA regulates the signal pathway by inhibiting the phosphatase activity of PP2C protein through its receptor PYL protein family, and PP2C and SnRKs were upregulated after drought stress. In our study, we found that genes related to the PYL4 receptor were inhibited by water stress, while PP2C and SnRKs were overexpressed.

Transcription factors (TFs) are proteins that bind to specific DNA sequences in order to regulate gene expression through their activation or repression [31]. Ref. [32] has reported numerous studies where the expression of transcription factors is closely related to the response to various abiotic factors such as drought acting on signal transduction pathways. In our study, we also found numerous transcription factors that were differentially expressed under stress conditions such as ethylene-responsive transcription factors, bZIP, WRKY and bHLH. This is in agreement with previous findings identifying several transcription factors involved in plant responses to drought by [33].

The expression of several stress-responsive genes is mediated by MYB family transcription factors, which are also involved in the ABA-dependent response, leading to the accumulation of ABA in cells [34]. MYB transcription factors are formed by one, two or three imperfect helix-turn-helix repeats and are grouped into three families depending on the MYB domain arrangement. The most common in plants is the R2R3 type, which has been described in potato as 123 MYB-like TFs (http://planttfdb.cbi.pku.edu.cn, accessed on 22 August 2022). WRKY-type transcription factors play an important role in plant response to abiotic factors and are activators of ABA signaling [35]. When we compared water-deficient plants with normally irrigated plants, we found that most genes related to WRKY transcription factors were downregulated under stress conditions.

According to [36], AtHB-7 is involved in plant stress tolerance and is dramatically upregulated after drought-stress treatment. In our case, overexpression of this gene was also found in both varieties when comparing drought-stressed plants (LOC102589092 and LOC102585726). These authors also demonstrated the function of StMYB1R-1 as a transcription factor, and its overexpression in transgenic potato enhanced the expression of drought-regulated genes such as AtHB-7, RD28, ALDH22a1 and ERD1-like, and improved plant tolerance to drought stress.

Phosphorylation and dephosphorylation of proteins regulated by kinases and phosphatases, respectively, is an effective mechanism in numerous signal transduction pathways. For example, MAPKs and CDPKs are known for their roles in water-stress signaling pathways. At the end of the phosphorylation cascade, transcription factors are either activated or suppressed by kinases or phosphatases regulating gene expression [37]. In our study, we also observed that in terms of molecular functions, genes related to hydrolase, kinase and phosphotransferase activity are significantly responsive under water restriction.

In the GO annotations of stressed and irrigated plants, DEG enrichment was observed in the photosystem-related category in the Agria variety, while the Zorba variety was not significant. The DEGs found in that GO were downregulated, as in the study presented by [25], demonstrating that under water-stress conditions, there is a decrease in photosynthesis, which helps to maintain the water status of the plant by inhibiting water uptake and plant growth. In addition, inhibition of photosynthesis-related genes leads to a decrease in stomatal conductance, reducing water loss through transpiration. These results are in agreement with the results of physiological data obtained for the same plants [38]. Potato leaf development is particularly sensitive to water stress. Drought first causes stomatal closure, reducing CO_2_ uptake for photosynthesis, reducing plant growth and yield.

Heat shock proteins are chaperones that contribute to protein stability under stress [39]. In our study, we observed differential expression of some heat shock proteins, particularly in Agria. Almost all of them were upregulated. Our results were similar to those obtained by [6] which suggest that there is a role for heat shock proteins in the maintenance of cell function under stress. Additionally, [24] reported an increased expression of heat shock proteins under water-stress conditions.

In both varieties, the most overexpressed gene under drought conditions was the “Abscisic acid and environmental stress-inducible protein TAS14-like”. In the study carried out by [40], it was also found that there was a significant correlation between TAS14 expression and early response to drought for recovering after stress. Increased TAS14 expression at the beginning of stress reduces the rate of photosynthesis, allowing rapid recovery of the plant’s water status. TAS14 protein is induced by abscisic acid and could act as a biomarker to evaluate the level of water stress in potato [41,42].

## 4. Materials and Methods

### 4.1. Plant Material and Growth Conditions

The potato varieties Agria and Zorba were used in this study. The trial was conducted in a greenhouse under optimal conditions for potato growing (18–22 °C, around 80% humidity) and a 16:8 h day–night light cycle. Eight tubers per variety and treatment were planted in 5 L pots with peat. Each pot was watered weekly with 1 L of water up to water holding capacity. The drought stress was applied 36 days after planting (DAP) and was maintained for 25 days without any water supply. Control plants were watered as normal [36]. After the drought period (61 DAP), samples for RNA extraction were collected from five plants of each variety and treatment. All leaf samples were immediately frozen in liquid nitrogen and stored at −80 °C until RNA extraction.

### 4.2. RNA Extraction and cDNA Library Preparation for Sequencing

Total RNA of leaf tissues from five biological replicates from each cultivar and treatment (20 samples in total) were isolated using the innuPREP Plant RNA kit (Analytik Jena GmbH, Jena, Germany) according to the manufacturer’s protocol. RNA quantification and quality of total RNA was measured using an Agilent 2100 bioanalyzer (Agilent Technologies, Santa Clara, CA, United States). The construction of the cDNA libraries using TruSeq stranded mRNA (Illumina, San Diego, CA, United States) and sequencing on a Novaseq 6000 150PE platform were performed by Macrogen Inc. (Seoul, Korea). These RNAseq data can be accessed at NCBI through SRA with the accession number PRJNA897005 (https://www.ncbi.nlm.nih.gov/sra/PRJNA897005, accessed on 3 November 2022).

### 4.3. Transcriptome Analysis

Quality control of the raw reads in each of the 20 libraries was performed by calculating the overall reads’ quality, total bases, total reads, GC (%) and basic statistics using fastqc. The trimming tool Trimmomatic [43] was used to remove adapter sequences and bases with base quality lower than three from the ends. Using the sliding window method, bases of reads that do not qualify for window size 4, and mean quality 15, were also removed. Afterwards, reads with length shorter than 36bp were dropped.

In order to map cDNA fragments obtained from RNA sequencing, trimmed reads were mapped to the reference genome GCF_000226072 using HISAT2 [44]. Known genes and transcripts were assembled with StringTie [45] based on the reference genome model. After assembly, the abundances of gene/transcripts were calculated as read counts, and FPKM (fragment per kilobase of transcript per million mapped reads) values or RPKM (reads per kilobase of transcript per million mapped reads) were used as normalization values. For the differential expression analysis between control and drought-stressed plants for both varieties, DESeq2 software [46] was applied, with q-value < 0.05 and fold change |log2| ≥ 2 as screening cutoffs.

### 4.4. GO Enrichment Analysis

GO (gene ontology) enrichment analysis of the DEGs was performed using the g:Profiler [47] tool. This tool performs statistical enrichment analysis to find over-representation of information from gene ontology terms, biological pathways, regulatory DNA elements, human disease gene annotations, and protein–protein interaction networks. The gene or gene product molecule associated with GO ID was summarized by parsing the ontology file and the annotation file for the GO graph structure.

### 4.5. Validation of Differentially Expressed Genes

In order to validate the RNAseq results, RT-qPCR was performed for twelve randomly selected DEGs; eight of these genes were common to both studied varieties and two others were variety-specific in each case. Specific primer pairs for the selected genes were designed with Primer3 software [48]. Their sequences are available in Appendix A. The β-tubulin gene was used as reference gene [49]. The RT-qPCR experiments were performed using a Roche LC480 II System (Roche Diagnostics Nederland BV, Almere, the Netherlands), with three technical and five biological replicates of the same samples used for RNAseq. Each PCR reaction contained 50 ng of cDNA, 100 nM of each primer and 1X PyroTaq EvaGreen qPCR Mix (CMB Cultek Molecular Bioline) and was adjusted with RNAse-free water to a final volume of 20 μL. The reactions were performed under the following conditions: 95 °C for 15 min, followed by 50 cycles at 95 °C for 15 s, 60 °C for 20 s and 72 °C for 20 s. For calculating and calibrating the expression levels of target genes in different varieties, the 2^−ΔΔCt^ method was applied [50].

## 5. Conclusions

In this study, we used the RNAseq technique to detect expression differences for better understanding the molecular mechanisms of tolerance to water stress in potato plants. Despite the difficulty of the study due to the fact that potatoes are tetraploid and highly heterozygous, numerous DEGs were found between stressed and control plants, showing that the plants have activated stress response mechanisms. The Agria variety showed significantly more differentially expressed genes, 931 upregulated and 2077 downregulated, compared to 735 upregulated and 923 downregulated genes in the Zorba variety, suggesting that Agria has a better response to lack of water.

We have identified the main DEGs and mechanisms regulating plant tolerance to water stress, such as some ABA-responsive genes, numerous transcription factors or heat shock proteins. We have also identified metabolic pathways involved in plant protective functions such as cell wall maintenance, carbohydrate metabolism or osmotic adjustment. These results suggest that plants have responded to stress as expected by activating stress-responsive metabolic pathways. These data provide the basis for the study of gene function and the mechanisms of regulation of tolerance to water stress. Due to the complexity of this trait in which numerous genes are involved, further studies are needed to assess the degree of contribution to tolerance of these genes and the identified metabolic pathways. One of the factors affecting drought response in plants is the duration and timing of stress application, so a comparison of DEGs at different times could be made in future studies.

## Figures and Tables

**Figure 1 ijms-23-13905-f001:**
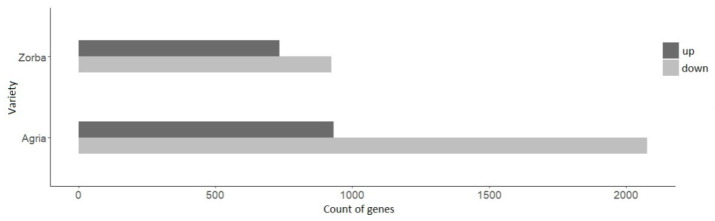
Count of genes up- and downregulated under drought stress in Agria and Zorba varieties.

**Figure 2 ijms-23-13905-f002:**
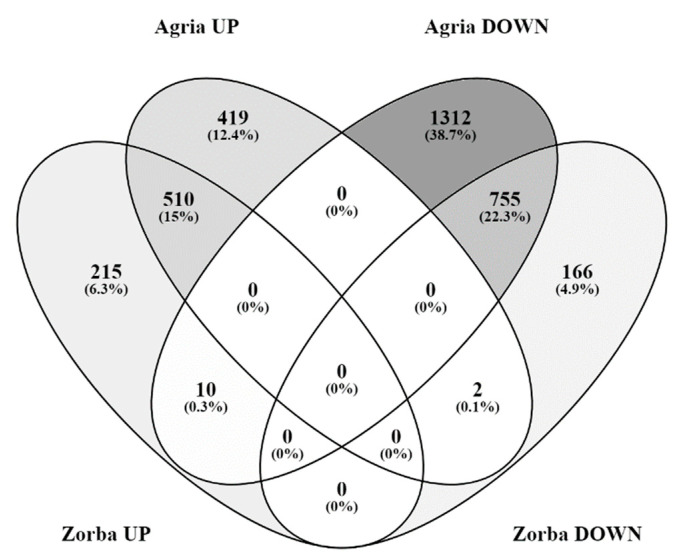
Venn diagram summarizing DEGs in the Agria and Zorba varieties in response to control and drought stress.

**Figure 3 ijms-23-13905-f003:**
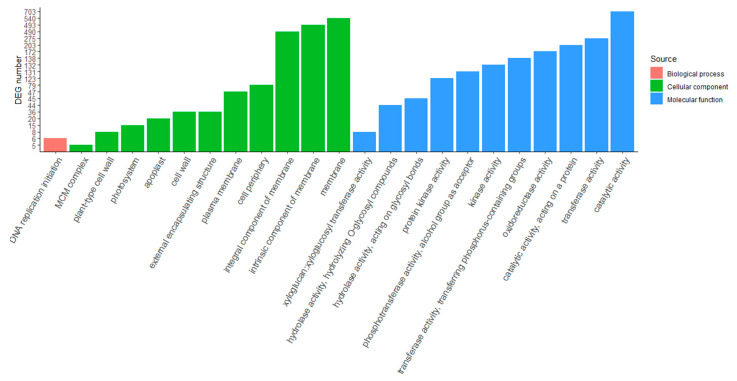
Gene ontology classification analysis of DEGs between control and drought-stressed potato leaves in Agria variety.

**Figure 4 ijms-23-13905-f004:**
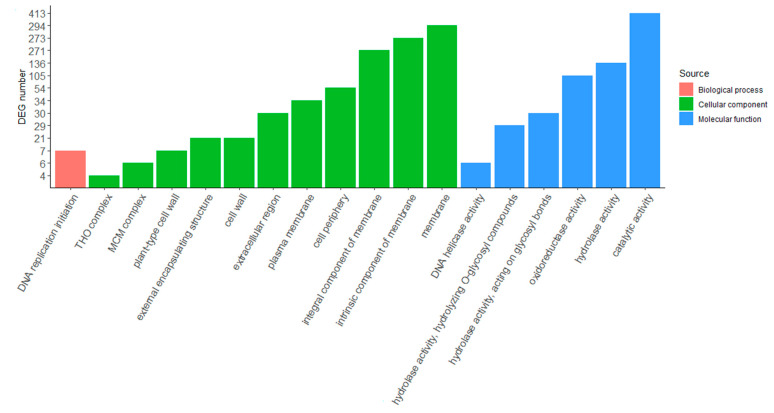
Gene ontology classification analysis of DEGs between control and drought-stressed potato leaves in Zorba variety.

**Figure 5 ijms-23-13905-f005:**
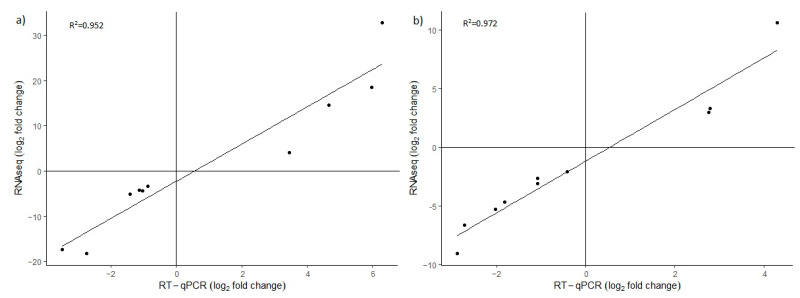
Validation of RNAseq results with RT-qPCR in (**a**) Agria variety and (**b**) Zorba variety.

**Table 1 ijms-23-13905-t001:** Summary of sequencing data.

		Control	Drought
Agria	Raw reads	57,004,343	54,509,300
	Clean reads	56,473,479	54,042,913
	Total mapped	50,569,589 (89.54%)	48,485,736 (89.71%)
	GC content (%)	43.63	43.38
	Q30 (%)	96.08	96.18
Zorba	Raw reads	56,278,132	53,964,119
	Clean reads	55,798,228	53,513,496
	Total mapped	50,271,909 (90.09%)	48,158,218 (89.99%)
	GC content (%)	43.48	43.41
	Q30 (%)	96.12	96.14

**Table 2 ijms-23-13905-t002:** List of genes showing the highest significantly different expression in the comparison of Agria control and drought-stressed plants.

Gene Name	Description	log2^fold change^	*p*-Value
Downregulated genes			
LOC102603621	-	−126.51	4.56 × 10^−7^
LOC102594756	36.4 kDa proline-rich protein-like	−39.22	5.85 × 10^−6^
LOC102596805	fasciclin-like arabinogalactan protein 2	−37.45	3.65 × 10^−8^
LOC102604005	putative calcium-binding protein CML19	−32.55	1.62 × 10^−4^
LOC102602308	hyoscyamine 6-dioxygenase	−31.44	3.14 × 10^−4^
LOC102582168	L-ascorbate oxidase-like	−30.82	5.63 × 10^−7^
LOC102603929	probable xyloglucan endotransglucosylase/hydrolase 1	−30.46	1.54 × 10^−4^
LOC102586959	probable WRKY transcription factor 53	−30.40	5.99 × 10^−7^
LOC102583042	1-aminocyclopropane-1-carboxylate synthase-like	−28.26	1.62 × 10^−7^
LOC102606325	GDSL esterase/lipase At5g33370-like	−27.13	3.40 × 10^−8^
Upregulated genes			
LOC107057685	abscisic acid and environmental stress-inducible protein TAS14-like	3449.52	1.28 × 10^−19^
LOC102583792	abscisic stress-ripening protein 2	149.50	2.89 × 10^−13^
LOC102606049	fidgetin-like protein 1	140.15	6.90 × 10^−10^
LOC102598218	translocator protein homolog	109.83	6.19 × 10^−18^
LOC102590433	-	73.36	5.58 × 10^−9^
LOC102606174	bZIP transcription factor 53-like	50.69	2.22 × 10^−15^
LOC102598306	SNF1-related protein kinase regulatory subunit gamma-like PV42a	48.76	3.16 × 10^−14^
LOC102592988	-	48.09	2.74 × 10^−16^
LOC102584616	-	42.46	2.71 × 10^−13^
LOC102591763	branched-chain-amino-acid aminotransferase 2, chloroplastic-like	39.31	1.50 × 10^−8^

**Table 3 ijms-23-13905-t003:** List of genes showing highest significantly different expression in the comparison of Zorba control and drought-stressed plants.

Gene Name	Description	log2^fold change^	*p*-Value
Downregulated genes			
LOC102598924	protein PMR5	−19.17	8.26 × 10^−18^
LOC102592481	-	−16.05	7.36 × 10^−13^
LOC102594756	36.4 kDa proline-rich protein-like	−15.42	1.85 × 10^−3^
LOC102599076	beta-hexosaminidase 3	−11.63	9.22 × 10^−10^
LOC102587252	protein STRICTOSIDINE SYNTHASE-LIKE 11-like	−11.33	5.95 × 10^−9^
LOC102605226	glucan endo-1,3-beta-glucosidase, basic isoform 1-like	−11.33	5.66 × 10^−8^
LOC102605560	glucan endo-1,3-beta-glucosidase, basic isoform 1	−10.91	5.85 × 10^−7^
LOC102590679	tropinone reductase homolog	−10.66	7.47 × 10^−14^
LOC102594482	delta(7)-sterol-C5(6)-desaturase-like	−10.37	2.30 × 10^−6^
LOC102592844	peroxidase 51	−10.31	1.63 × 10^−15^
Upregulated genes			
LOC107057685	abscisic acid and environmental stress-inducible protein TAS14-like	692.09	1.47 × 10^−12^
LOC102606049	fidgetin-like protein 1	177.36	2.71 × 10^−10^
LOC102590433	-	140.98	8.29 × 10^−11^
LOC102580665	O-acyltransferase WSD1-like	69.42	4.71 × 10^−22^
LOC102577501	nonspecific lipid transfer protein a7	53.62	2.18 × 10^−9^
LOC102596984	nonspecific lipid-transfer protein 2-like	50.35	1.77 × 10^−8^
LOC102597309	nonspecific lipid-transfer protein 2-like	48.98	5.38 × 10^−11^
LOC102598306	SNF1-related protein kinase regulatory subunit gamma-like PV42a	48.30	2.75 × 10^−13^
LOC102582408	probable protein phosphatase 2C 51	37.80	5.86 × 10^−10^
LOC102587411	MLO-like protein 6	36.71	7.14 × 10^−6^

## Data Availability

Not applicable.

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
