# Peer review of "Transcriptome Analysis of Two Tetraploid Potato Varieties under Water-Stress Conditions"

_ijms, 2022, doi:10.3390/ijms232213905_

Round 1

Reviewer 1 Report

In the reviewed manuscript, the authors carried out the expression analysis using the RNA seq between control and water-stressed plants in varieties named Agria and Zorba. Agria variety, it was found that 931 genes were up-regulated and 2077 genes were down-regulated under stress conditions. When both treatments were compared in Zorba plants, 735 genes were found to be up-regulated and 923 genes were found to be down-regulated. Significantly, more DEGs were found in Agria variety, indicating a good stress response of this variety. "Abscisic acid and environmental stress inducible protein TAS14 like" was the most overexpressed gene under drought conditions in both varieties, but expression differences were also found in numerous transcription factors or heat shock proteins. The principal GO term found was “cellular components”, more specifically related to the cell membrane and the cell wall. These results provide valuable information related to the molecular mechanisms of tolerance to water stress in order to establish the basis for breeding new more tolerant varieties. The subject of the manuscript is topical, very significant and interesting. The study is well-conducted and provided important results that might use to improve drought tolerance in plants. Overall, the amount of work is very less and the data are short and just does not meet the criteria of publication in current form (need more figures and supplementary data). However, major revisions are suggested as shown below;

  1. The manuscript has a few mistakes in English grammar and sentence structure, so I recommend the authors to used an English proofreading service.
  2. The abstract should highlight the most important results of the parameters and characteristics assayed.
  3. The introduction looks shallow, write more detail about the studied issue.
  4. General note: the figures in this section are quite low resolution and difficult to make out. Higher-resolution versions will be needed for publication, for example, in Figures 2, 3, and 4.
  5. Why you selected Agria and Zorba. varieties for your experiment
  6. In Material and Methods:- indicate how many replicates assayed in each analysis/parameter. The number of sample or biological and technical replicates should be mentioned for each parameter in the methods
  7. Results are explained in detail.
  8. Why authors only selected the 12 genes for validate the RNA seq results
  9. The discussion should be interpreted with the results as well as discussed in relation to the present literature.
  10. The conclusion section is very poorly written. It should be extensively improved.
  11. References: shall have to correct the whole References according to the ”Instructions for the Authors”, e.g. the Journal name is in italics, in the case of the publication year shall have to use bold numbers, the volume is in italics and you shall have to use the abbreviated number of the Journals cited.

Reviewer 2 Report

I had a great opportunity to review manuscript entitled: 'Transcriptome analysis of two tetraploid potato varieties under water stress conditions' which is considered for publication in International Journal of Molecular Sciences. The goal of this manuscript is to present and characteristics of changes in the transcriptome in two tetraploid potato varieties under water stress conditions. The manuscript is elaborated on an interesting topic.

GENERAL COMMENTS:
TITLE
The paper title is well stated, it is informative and concise.

ABSTRACT, INTRODUCTION
Abstract is well written with the key findings of the study. Introduction is concise, focused and informative.

MATERIAL AND METHODS
Material and research methods are presented appropriately and clearly. Experimental setup and the description in the methods section are well structured, and the statistical analysis is done alright. In spite of that I have a one objections against its present form:
- RNA-seq data should be archived in NCBI. Where is the RNA-seq data deposited?

RESULTS
The results obtained in this study are interesting. Results presented correctly.

DISCUSSION and CONCLUSIONS
In general, the discussion of results is correct and sufficient.
Please focus on the findings from the results in this research and finally answer the scientific problem.
In the Conclusion I suggest to writing two other words on the aspect concerning the aspects where the future studies must be oriented.

LITERATURE
The items of literature included in the paper are rather sufficient and adequate to the subject of the paper. If software or online applications were used in the work, it is advisable to include references!

The text of the manusctipt is not formatted correctly yet.
I also recommend revision of the English language by a native speaker or a commercial entity to remove minor typos in the text.

Round 2

Reviewer 1 Report

Dear Editor,

Thank you for providing the opportunity to review the revised manuscript. The manuscript is improved considerably after revision according to the reviewer's comment. Now this study is a suitable contribution to the IJMS. I recommend the manuscript for publication.

Thank you

With best regards